# Effects of Sacubitril/Valsartan on the Right Ventricular Arterial Coupling in Patients with Heart Failure with Reduced Ejection Fraction

**DOI:** 10.3390/jcm9103159

**Published:** 2020-09-29

**Authors:** Daniele Masarone, Vittoria Errigo, Enrico Melillo, Fabio Valente, Rita Gravino, Marina Verrengia, Ernesto Ammendola, Rossella Vastarella, Giuseppe Pacileo

**Affiliations:** Heart Failure Unit, Department of Cardiology, AORN dei Colli-Monaldi Hospital, 80121 Naples, Italy; vittoria.errigo@gmail.com (V.E.); enrico.melillo@ospedalideicolli.it (E.M.); fabio.valente@ospedalideicolli.it (F.V.); rita.gravino@ospedalideicolli.it (R.G.); marina.verrengia@ospedalideicolli.it (M.V.); ernesto.ammendola@ospedalideicolli.it (E.A.); rossella.vastarella@ospedalideicolli.it (R.V.); giuseppe.pacileo@ospedalideicolli.it (G.P.)

**Keywords:** sacubitril/valsartan, heart failure reduced ejection fraction, right ventricle, right ventricular arterial coupling

## Abstract

Background: right ventricle-pulmonary artery (RV-PA) coupling assessed by measuring the tricuspid anular plane systolic excursion (TAPSE)/pulmonary artery systolic pressure (PASP) ratio has been recently proposed as an early marker of right ventricular dysfunction in patients with heart failure with a reduced ejection fraction (HFrEF). Methods: As the effects of sacubitril/valsartan therapy on RV-PA coupling remain unknown, this study aimed to analyse the effect of this drug on TAPSE/PASP in patients with HFrEF. We retrospectively analysed all outpatients with HFrEF referred to our unit between October 2016 and July 2018. Results: At the 1-year follow-up, sacubitril/valsartan therapy was associated with a significant improvement in TAPSE (18.26 ± 3.7 vs. 19.6 ± 4.2 mm, *p* < 0.01), PASP (38.3 ± 15.7 vs. 33.7 ± 13.6, *p* < 0.05), and RV-PA coupling (0.57 ± 0.25 vs. 0.68 ± 0.30 *p* < 0.01). These improvements persisted at the 2-year follow-up. In the multivariable analysis, the improvement in the RV-PA coupling was independent of the left ventricular remodelling. Conclusions: in patients with HFrEF, sacubitril/valsartan improved the RV-PA coupling; however, further trials are necessary to evaluate the role of sacubitril/valsartan in the treatment of right ventricle (RV) dysfunction either associated or not associated with left ventricular dysfunction.

## 1. Introduction

Sacubitril/valsartan is the latest “disease-modifying drug” approved for the management of patients with heart failure with reduced ejection fraction (HFrEF) [1].

In addition to a reduction in the rate of mortality and hospitalisation [2,3], both randomised clinical trials and real-life studies have shown that sacubitril/valsartan induced the “reverse remodelling” of the left ventricle (LV), with a reduction in the ventricular volumes, an increase in the ejection fraction (EF) [4,5], an improvement in the diastolic function [6,7], and a reduction in the degree of functional mitral regurgitation [8]. In patients with HFrEF, the increased LV filling pressure induces right ventricle (RV) chronic overload, which increases RV afterload and thus leads to RV remodelling, with a reduction in the performance of RV even when it is not directly involved in the development of cardiovascular disease [9,10].

Despite the considerable amount of data on the benefits of sacubitril/valsartan for LV function, data on the effect of this drug on the RV are limited. RV-pulmonary artery (RV-PA) coupling, an index of the in vivo RV length vs. developed force, and a non-invasive assay of the RV contractile state [11] have been proposed as early markers of RV ventricular dysfunction [12]. Recently, the tricuspid annular plane systolic excursion (TAPSE)/pulmonary artery systolic pressure (PASP) ratio has been proposed as the best echocardiographic method to evaluate it [13]. In patients with HFrEF, the effects of sacubitril/valsartan therapy on RV-PA coupling remain unknown; thus, we aimed to analyse the effect of sacubitril/valsartan on the TAPSE/PASP ratio in this patient group.

## 2. Methods

We retrospectively analysed all outpatients with HFrEF, defined as those with an EF ≤40%, referred to the Heart Failure Unit of Ospedale Monaldi (Naples, Italy), between October 2016 and July 2018. This study was conducted in accordance with the Declaration of Helsinki, all the patients provided written informed consent and the study was approved by local ethical committee (455 of June 2020).

We excluded patients with severe renal disease (i.e., estimated glomerular filtration rate according to Chronic Kidney Disease Epidemiology Collaboration (CKD-EPI) equation ≤30 mL/min/1.73 m^2^), moderate-severe liver disease (i.e., a Child-Pugh score ≥7), severe chronic obstructive pulmonary disease (i.e., Global Initiative for Chronic Obstructive Lung Disease class ≥3), prior heart surgery with pericardial incision, prior pulmonary embolism, and HF-related hospitalisation ≤3 months before the administration of angiotensin receptor neprilysin inhibitor.

During the study period, 215 patients with HFrEF received sacubitril/valsartan therapy; among these patients, 8 (3.7%) with severe renal disease, 7 (3.2%) with moderate-severe liver disease, 9 (4.1%) with moderate-severe chronic obstructive pulmonary disease, 7 (3.2%) who previously underwent heart surgery with pericardial incision, and 21 (9.7%) who were recently hospitalised due to HF were excluded.

In the 163 patients enrolled in the study, sacubitril/valsartan was administered at a dose of 24/26 mg twice daily in 102 patients (62.5%) and at a dose of 49/51 mg twice daily in the remaining 61 patients (37.5%). A complete echocardiographic examination was performed in the echocardiography laboratory by an experienced and skilled cardiologist.

The mean pulmonary artery pressure (mPAP) was calculated as 0.61 × PASP + 2.

RV-PA coupling, estimated as the TAPSE/PASP ratio values, was detected at the beginning of the sacubitril/valsartan therapy, and those measured at 1 year and at 2 years after the initiation of therapy were evaluated.

All the statistical analyses were performed using Prism 8 (GraphPad Software, San Diego, CA, USA). Variables with a normal and non-normal distribution were expressed as mean  ±  standard deviation and median/interquartile range, respectively. The differences between the baseline and treatment were compared using the Wilcoxon signed-rank test for variables with a non-normal distribution and the t-test for dependent samples for variables with a normal distribution. All the *p*-values were two-sided, and a *p*-value of <0.05 was considered significant.

## 3. Results

The demographic, clinical, and echocardiographic characteristics of the overall population are shown in Table 1.

The patients’ mean age was 57.9 ± 12.3 years, and 78.1% of the population were men. Approximately 31.3% of the patients had New York Heart Association (NYHA) class III.

Of the total population, 83 (50.9%) had ischemic dilated cardiomyopathy according to Felker’s definition [14].

Hypertension was noted in 62.5% of the patients, ischaemic heart disease in 50.9%, diabetes in 48.4%, and chronic obstructive pulmonary disease in 37.4%. The mean left ventricular ejection fraction (LVEF) was 28.9% ± 6.4%.

All the patients were treated with β-blockers, 18.4% with ivabradine, 64.4% with mineralocorticoid receptor inhibitors, and 23.3% with digoxin. Of the diabetic patients, 34 (20,8%) had SGLT2 inhibitors.

During follow-up sacubitril/valsartan was up-titrated according to standard practice and at two years 115 patients (70.5%) took 97/103 mg twice daily, 40 (24.5%) patients the dose of 49/51 mg twice daily and 8 patients (5%) the dose of 24/26 mg twice daily.

With the exception of a statistically significant reduction in the dose of furosemide (75 ± 25 mg vs. 50 ± 12.5 mg; *p* < 0.01), no significant changes were observed in the other “disease-modifying” drugs during follow-up.

The improvement of the major echocardiographic parameters after the initiation of sacubitril/valsartan is summarised in Table 2.

At the 2-year follow-up, sacubitril/valsartan therapy was associated with a significant improvement in a series of left-side echo parameters: the LVEF (28.9% ± 6.4% vs. 33.4% ± 4.8%, *p* < 0.01), LV end diastolic volume (237.2 ± 87.6 mL vs. 208.4 ± 52.4 mL, *p* < 0.05), LV end-systolic volume (179.5 ± 65.3 vs. 157.9 ± 45.2 mL, *p* < 0.05), left atrial volume index (37.6 + 5.2 vs. 31.8 + 3.9, *p* < 0.01), and estimated mean pulmonary capillary wedge pressure (22.1 ± 7.7 vs. 19.8 ± 6.8, *p* < 0.05).

With regard to the RV function (Figure 1), there was a significant improvement in the TAPSE (18.76 ± 3.7 vs. 19.6 ± 6.8 mm, *p* < 0.01), peak systolic S wave (10.4 ± 3.2 vs. 11.2 ± 2.9, *p* < 0.05), PASP (38.3 ± 15.7 vs. 27.3 ± 13.6, *p* < 0.05), mPAP (24.1 ± 12.6 mmHg vs. 20.8 ± 11.3, *p* < 0.05), and RV-PA coupling (0.57 ± 0.25 vs. 0.68 ± 0.30 *p* < 0.01) at the 1-year follow-up.

The improvements in RV-PA coupling persisted at the 2-year follow-up (Figure 2 and Figure 3).

In the subgroup analysis, the effects of sacubitril/valsartan on the RV-PA coupling were the same in men and women (Δ RV-PA 0.14 ± 0.09 vs. 0.15 ± 0.03; *p* = 0.98), in diabetic and non-diabetic patients (Δ RV-PA 0.15 ± 0.01 vs. 0.16 ± 0.05; *p* = 0.09, and in non-ischemic patients when compared with ischemic patients (Δ RV-PA 0.16 ± 0.08 vs. 0.15 ± 0.07; *p* = 0.07).

The effects on RV-PA arterial coupling was higher in patients that assumed the dose of 97/103 twice daily respect patients that assumed the dose of 49/51 mg twice daily (Δ RV-PA 0.16 ± 0.07 vs. 0.14 ± 0.05; *p* = < 0.05).

In the multivariable analysis, the improvement in the RV-PA coupling was independent of the LV reverse remodelling; however, it was correlated with a reduction in the left atrial volume index (Table 3).

## 4. Discussion

To the best of our knowledge, this is the first study to report an improvement in RV-PA coupling after sacubitril/valsartan therapy in a real-world population. The main findings of this study are as follows: (1) sacubitril/valsartan improves RV-PA coupling, (2) this improvement is related to both an increase in the TAPSE and a reduction in the pulmonary arterial pressure, and (3) the improvement in the TAPSE/PASP ratio is independent of the LV reverse remodelling.

In HFrEF, a low TAPSE value (i.e., <16 mm) indicates an advanced disease stage and leads to an increased risk of death by HF, sudden death, and hospitalisation [15,16].

In patients with HFrEF, an increase in the left atrial (LA) pressure and a reduction in LA compliance [17,18] leads to LA remodelling (increase in LA size, impaired LA contractility, and interstitial fibrosis), resulting in an increase in LA stiffness, which is a major determinant of pulmonary hypertension [19]. In fact, an increase in the LAVI and left atrial pressure ensures that LA no longer acts as a barrier between the high left ventricular pressure and the pulmonary vessels, thus resulting in a passive transmission of the left ventricular pressure into the pulmonary vascular tree [20].

PASP was associated with a prognosis in HF with type II pulmonary hypertension [21,22].

RV-PA coupling quantifies the adaptation of the RV to its afterload and is considered to be one of the main determinants of functional capacity [23] and survival in patients with HF [24,25].

The gold standard in quantifying RV-PA coupling is the RV end-systolic elastance to the pulmonary arterial elastance ratio, measured invasively with multi-beat RV pressure-volume loop acquisitions by conducting a dedicated right heart catheterisation [26,27]. However, the routine use of this technique in clinical practice is not indicated, as it is invasive and requires specific conductance catheters.

Compared with right cardiac catheterisation, which is the reference method, two-dimensional echocardiography allows the calculation of combined indices such as the TAPSE/PASP ratio, which is a reliable index of RV-PA coupling [28,29]. The TAPSE/PASP ratio has important prognostic implications; in fact, a TAPSE/PASP ratio < 0.36 mm/mmHg identified patients with a high risk, irrespective of EF status [30].

To date, data on the effects of β-blockers and renin-angiotensin-aldosterone inhibitors on the RV function are limited [31,32,33,34], despite the significant results in determining the reverse remodelling of the LV. Recently, in a pressure overload model of pulmonary hypertension, sacubitril/valsartan, due to the combined natriuretic and vasodilator effects, prevent maladaptive RV remodelling via the amelioration of RV contractility and relaxation, reduction in RV afterload, and improvement in RV-PA coupling [35].

In a recent study enrolling 60 consecutive patients with HFrEF, Correale et al. showed that sacubitril/valsartan increased the TAPSE (7.8 ± 3.9 vs. 16.5 ± 4.0 mm, *p* < 0.001) and decreased the PASP (31.0 ± 12.8 vs. 34.7 ± 12.5 mmHg, *p* < 0.05) [36].

In our study, we evaluated the effects of sacubitril/valsartan on RV-PA coupling by measuring the TAPSE/PASP ratio; the results clearly indicate that sacubitril/valsartan improved the RV performance by acting on the RV contractility (indirectly estimated by TAPSE and S wave) and reducing the RV afterload (indirectly estimated by PASP), with an improvement in the RV-PA coupling.

The same improvements were observed in men and women, in diabetic patients compared with non-diabetic patients, and in ischemic patients compared with non-ischemic patients.

These findings are in agreement with those of previous studies, which showed that sacubitril/valsartan therapy resulted in an equal inverse remodelling of the LV in women, diabetic individuals, and ischemic heart disease patients [37,38,39].

Therefore, both the remodeling of the LV and the favourable effects on the function of the RV are independent of these parameters but are related to the dose of sacubitril/valsartan, confirming that higher the dose is related to better outcome. Consequently, any action must be taken to reach the target dose of sacubitril/valsartan or the maximum dosage tolerated by the patient.

Finally, the finding that the improvement in the RV function is not related to the reverse remodelling of LV but only to the reduction in LAVI is also of great interest; in fact, LAVI reflects the magnitude and chronicity of elevated cardiac filling pressures. Hence, the improvement in LAVI is indicative of a reduction in the LV filling pressure, which results in a reduction in pressure transmission to the pulmonary vascular shaft with a reduction in both the PASP and mPAP.

Future studies are required to confirm these data, which would then allow us to hypothesise the use of sacubitril/valsartan even in patients with heart failure (HF) and isolated/predominant RV dysfunction.

## 5. Study Limitation

The small sample size, single-centre study design, and observational nature of the study may affect the generalizability of our results. These preliminary results need to be confirmed in a properly powered multicentric study and randomised trials.

## 6. Conclusions

In patients with HFrEF, sacubitril/valsartan therapy increased the TAPSE and decreased the PASP with an improvement in the RV-PA coupling; this effect is not related to LV reverse remodelling. Further studies are necessary to confirm these data in order to evaluate the role of sacubitril/valsartan in the treatment of RV dysfunction in HFrEF patients.

## Figures and Tables

**Figure 1 jcm-09-03159-f001:**
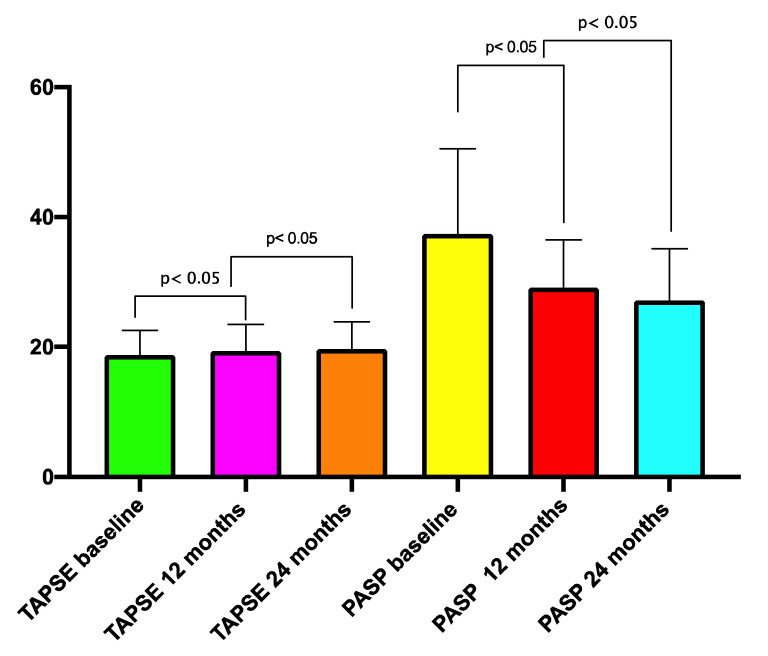
Changes in the TAPSE and PAPS during follow-up. TAPSE—tricuspid annular systolic pressure; PAPS—pulmonary artery systolic pressure.

**Figure 2 jcm-09-03159-f002:**
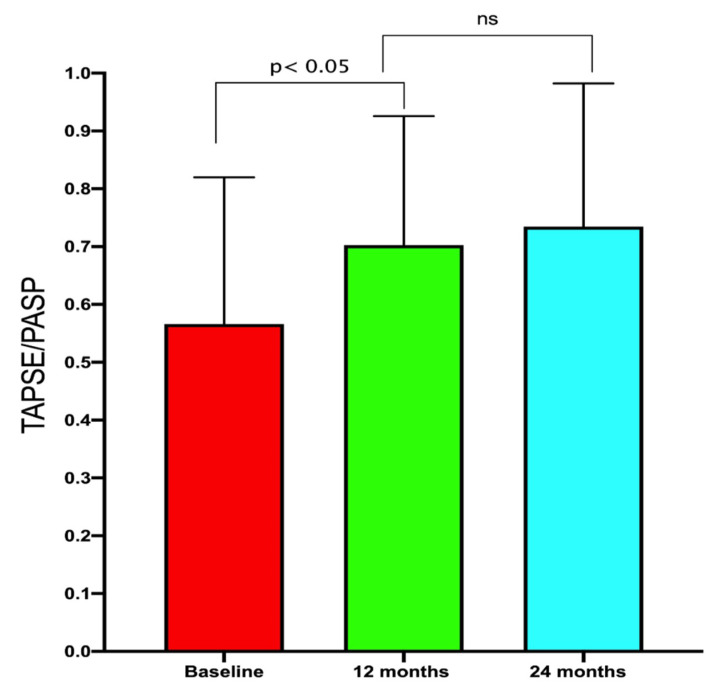
Improvement in the right ventricular arterial coupling during follow-up. TAPSE—tricuspid annular plane systolic excursion; PASP—pulmonary artery systolic pressure; ns—not significant.

**Figure 3 jcm-09-03159-f003:**
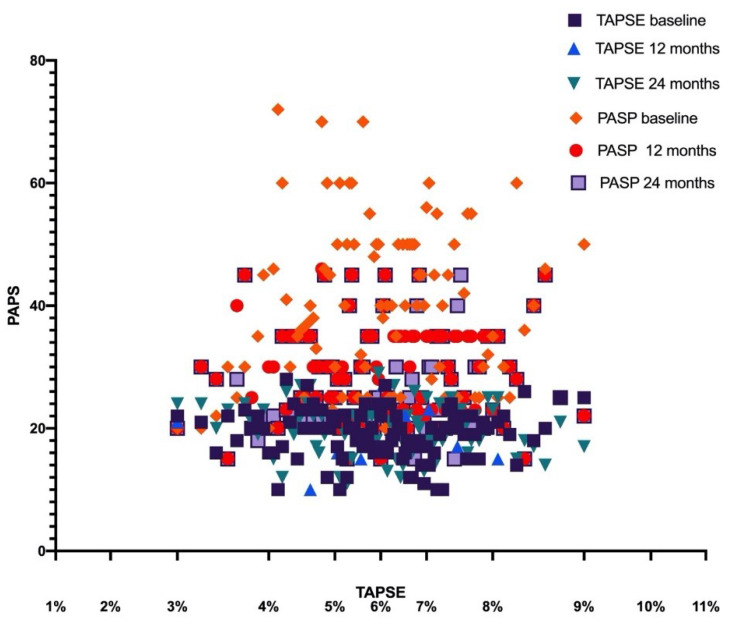
Scatterplot analysis of the PAPS and TAPSE at baseline and after sacubitril/valsartan therapy. TAPSE—tricuspid annular plane systolic excursion; PASP—pulmonary artery systolic pressure.

**Table 1 jcm-09-03159-t001:** Baseline characteristics of the overall population.

Variable	Overall Population (*n* = 163)
Age (mean ± SD)	57.9 ± 12.3 years
Female sex (*n*, %)	52 (31.9%)
Ischemic (*n*, %)	83 (50.9%)
Hypertension (*n*, %)	102 (62.5%)
Diabetes (*n*, %)	79 (48.4%)
COPD (*n*, %)	61(37.4%)
NYHA II (*n*, %)	112 (68.7%)
NYHA III (*n*, %)	51 (31.3%)
Systolic blood pressure (mean ± SD)	119 ± 14.8 mmHg
Diastolic blood pressure (mean ± SD)	72.7 ± 9.3 mmHg
Heart rate (mean ± SD)	68.2 ± 14.6 bpm
LVEDV (mean ± SD)	237.2 ± 87.6 mL
LVESV (mean ± SD)	179.5 ± 65.3 mL
LVEF (mean ± SD)	28.9 ± 6.4%
E/e’ average (mean ± SD)	14.5 ± 4.8 cm/sec
LAVI	37.6 ± 5.2 mL/m^2^
Creatinine (mean ± SD)	1.3 ± 1.1 mg/dL
e-GFR (mean ± SD)	63.6 ± 15.2 mL/min
NT-proBNP (mean ± SD)	1716 ± 954 pg/mL
Loop diuretic (*n*, %)	118 (72.3%)
Furosemide dose (mean ± SD)	75 ± 25 mg
Beta-blockers (*n*, %)	163 (100%)
Carvedilol dose (mean ± SD)	37.5 ± 612.5 mg
Bisoprolol dose (mean ± SD)	5 ± 3.75 mg
ACEi/ARBs (*n*, %)	158 (96.9%)
Ramipril dose (mean ± SD)	5 ± 3.75 mg
Valsartan dose (mean ± SD)	120 ± 80 mg
MRA (*n*, %)	105 (64.4%)
Eplerenone dose (mean ± SD)	25 ± 12.5 mg
Ivabradine (*n*, %)	30 (18.4%)
Ivabradine dose (mean ± SD)	10 ± 5 mg
Digoxin (*n*, %)	35 (23.3%)
Digoxin dose (mean ± SD)	0.09375 ± 0.0625 mg

SD—standard deviation; COPD—chronic obstructive pulmonary disease; NYHA—New York Heart Association; LVEDV—left ventricular end diastolic volume; LVESV—left ventricular end systolic volume; LVEF—left ventricular ejection fraction; E/e’—ratio of the transmitral early peak velocity E by pulsed wave Doppler over the early diastolic mitral annulus velocity; LAVI—left atrium volume index; e-GFR—the estimated glomerular filtration rate; NT-proBNP—N terminal pro brain natriuretic peptide; ACEi—angiotensin-converting enzyme inhibitors; ARBs—angiotensin receptor blockades; MRA—mineralocorticoid receptor antagonist.

**Table 2 jcm-09-03159-t002:** Changes in the echocardiographic parameters during follow-up.

Variable	Baseline	1-Year Follow-Up	*p*	2 Years Follow-Up	*p*
LVEDV (mean ± SD)	237.2 ± 87.6 mL	213.3 ± 64.8 mL	<0.05	208.4 + 52.4 mL	<0.05
LVESV (mean ± SD)	179.5 ± 65.3 mL	165.4 ± 52.7 mL	<0.05	157.9 ± 45.2 mL	<0.05
LVEF (mean ± SD)	28.9 ± 6.4%	31.5 ± 6.2%	<0.05	33.4% ± 4.8%	<0.01
LAVI (mean ± SD)	37.6 ± 5.2 mL/m^2^	34.1 ± 4.4 mL/m^2^	<0.01	31.8 ± 3.9 mL/m^2^	<0.01
TAPSE (mean ± SD)	18.76 ± 3.7 mm	19.3 ± 3.2 mm	<0.05	19.6 ± 6.8 mm	<0.05
mPAP (mean ± SD)	24.1 + 12.6 mmHg	22.7 ± 10.9 mmHg	<0.05	20.8 + 11.3 mmHg	<0.05
PASP (mean ± SD)	38.3 ± 15.7 mmHg	29.1 ± 14.8 mmHg	<0.01	27.3 ± 13.6 mmHg	<0.01

LVEDV—left ventricular end diastolic volume; LVESV—left ventricular end systolic volume; LVEF—left ventricular ejection fraction; E/e’—ratio of the transmitral early peak velocity E by pulsed wave Doppler over early diastolic mitral annulus velocity; LAVI—left atrium volume index; TAPSE—tricuspid anular plane systolic excursion; mPAP—medium pulmonary artery pressure; PASP—pulmonary artery systolic pressure.

**Table 3 jcm-09-03159-t003:** Multiple linear regression analysis on the Δ RV-PA coupling.

Variable	Mean + SD	Β	t	*p*
Δ RV-PA coupling	0.16 ± 0.03	-	-	-
Δ LVEDV	29.6 ± 15.8 mL	−0.058	0.190	0.849
Δ LVESV	21.68 ± 8.36 mL	0.017	0.386	0.700
Δ LVEF	4.5 ± 0.9%	0.186	0.391	0.697
Δ E/e’ average	3.58 ± 1.52 cm/s	−0.381	0.186	0.852
Δ LAVI	6.38 ± 2.5 mL/m^2^	3.075	0.2378	0.045
Δ NT-pro BNP	474.4 ± 254.6 pg/mL	−0.071	0.390	0.697
Δ mPAP	3.33 ± 1.65	0.014	0.869	0.993

Δ—difference between value at baseline and value at follow-up; RV-PA—right ventricle-pulmonary artery; LVEDV—left ventricular end diastolic volume; LVESV—left ventricular end systolic volume; LVEF—left ventricular ejection fraction; E/e’—ratio of the transmitral early peak velocity E by pulsed wave Doppler over early diastolic mitral annulus velocity; NT-proBNP—N terminal pro brain natriuretic peptide; LAVI—left atrial volume index; mPAP—mean pulmonary artery pressure.

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
