# Peer review of "Effects of Sacubitril/Valsartan on the Right Ventricular Arterial Coupling in Patients with Heart Failure with Reduced Ejection Fraction"

_jcm, 2020, doi:10.3390/jcm9103159_

Round 1

Reviewer 1 Report

In this fascinating manuscript Polito et al set out to retrospectively evaluate the effect of sacubitril/ valsartan on echocardiographic parameters at one and two years of the therapy in an ambulatory heart failure population.    The authors reviewed the echocardiograms of all out-patients with HFrEF and for those were initiated on sacubitril/valsartan they compared the echocardiograms prior to beginning entresto and 1 year afterwards. Their findings demonstrated no significant differences in multiple left ventricular parameters including, but not limited to LVEDV, LVESV or LVEF. There was also no statistically significant difference in NT-proBNP or E/e’ PA mean pressure at 1 year.   There were, however, statistically significant differences in PASP and TAPSE at 1 year and 2 years, despite there being no statistically significant difference in left ventricular parameters. The decline in PASP and TAPSE is intriguing and the clinical relevance is clear.  It is a bit of a challenge to reconcile the lack of significant change in the LV parameters given the well documented effect of sacubitril/valsartan on LV stroke volume, ejection fraction, E/A and natriuretic peptides as demonstrated in multiple other publications.

There are several elements that might help with the interpretation of this data. First, a more detailed description of the baseline characteristics of the population. The ongoing treatments at baseline and at the time of the follow-up measurements (diuretics, other vasoactive agents) would also be useful as this information is necessary in order to delineate how much of the differences seen are truly attributable to sacubitril/valsartan. Given that the entire population was measured evaluation of the non-sacubitiril/valsartan patients.  It must be noted that even with this information it will be difficult to come to this conclusion in the absence of information on right-sided filling or to reconcile the lack of changes to left-sided function.

In addition, in Table 2 the change in PASP and PAMP are shown as positive although the values decrease and should be represented as negative change given that both decrease.

Author Response

We thank the reviewer for the helpful comments. However, in our population, treatment with sacubitril/valsartan improved the echocardiographic parameters of both the left ventricle (LVEDV reduction, LVESV, EF increase, NT-proBNP decrease, and E/e' average reduction) and the right ventricle (increase in TAPSE and reduction in PASP and mPAP). In the multivariate analysis, the improvement in the RV–PA coupling did not show any correlations with the reverse remodeling of the left ventricle. In accordance with the reviewer’s suggestion, we expanded the baseline population characteristics and specified that there were no significant changes in the dosage of other disease modifiers.

According to other reviewer suggestions, PASP and TAPSE were removed from the multiple linear regression analysis.

Reviewer 2 Report

In this study, it was found that the administration of sacubitril/valsartan to patients with HFrEF improved mPAP/TAPSE after 12 and 24 months. Although this is an one-arm observational study, it is an interesting result and is expected to provide useful information to the reader. The following are my comments.

  • Please specify the definition of HFrEF used in this study and the underlying disease of the patient (ischemic cardiomyopathy? dilated cardiomyopathy?...).
  • In Table 1 and Table 2, the decimal point should be "." instead of ",".
  • Table 2: Some indicators have too many decimal places.
  • It is better to present the values of major parameters such as SPAP, TAPSE and E/e’ at baseline, ​​one year and two years in a table.
  • For TAPSE and SPAP, why not add a bar graph like figure 1?
  • In terms of action of sacubitril / valsartan, it is easy to think that the left atrial pressure decreased due to some mechanism, resulting in lower pulmonary arterial pressure and improved TAPSE. However, the change in E/e', which reflects left atrial pressure, was not significant. I would like to ask you to consider this point.
  • In the multiple linear regression analysis on the change of RV-PA coupling (Table 2), PASP and TAPSE are included in the factors, but since those are terms of the calculation formula of RV-PA coupling, it seems natural to be related.
  • Was there a connection between SPAP and TAPSE? Interesting results may be seen by plotting the values ​​at baseline, 1 year and 2 years for each case, with the vertical axis representing SPAP and the horizontal axis representing TAPSE.

Author Response

  • Please specify the definition of HFrEF used in this study and the underlying disease of the patient (ischemic cardiomyopathy? dilated cardiomyopathy?...).

Response: We thank the reviewer for the valuable comments. We specified that of the total study population, 83 (50.9%) had ischemic dilated cardiomyopathy according to Felker’s definition.

  • In Table 1 and Table 2, the decimal point should be "." instead of ",".

Response: We thank the reviewer for his useful comment. The comma was changed to the decimal point in accordance with the reviewer’s suggestion.

  • Table 2: Some indicators have too many decimal places.

Response: We thank the reviewer for his helpful comment. We revised this part in accordance with the reviewer’s suggestion.

It is better to present the values of major parameters such as SPAP, TAPSE and E/e’ at baseline, ​​one year and two years in a table.

Response: We thank the reviewer for his valuable comment. We added Table 2 in accordance with the reviewer’s suggestion.

  • For TAPSE and SPAP, why not add a bar graph like figure 1?

Response: Thank you for clarifying. We added a table (Table 2) presenting the changes in the main echocardiographic parameters during follow-up.

  • In terms of action of sacubitril/valsartan, it is easy to think that the left atrial pressure decreased due to some mechanism, resulting in lower pulmonary arterial pressure and improved TAPSE. However, the change in E/e', which reflects left atrial pressure, was not significant. I would like to ask you to consider this point.

Response: We agree with the reviewer that E/e' should have been reduced after the sacubitril/valsartan therapy, considering the decrease in PASP. However, the large gray area of this parameter (in which the values between 8 and 15 cannot determine whether left ventricular filling pressures have increased or not) could explain this result. In the multivariate analysis, we included the LAVi (which also reflects the left atrial pressure) whose value decreases significantly during the follow-up and whose variation is associated with the improvement in the TAPSE-PASP ratio.

  • In the multiple linear regression analysis on the change of RV-PA coupling (Table 2), PASP and TAPSE are included in the factors, but since those are terms of the calculation formula of RV-PA coupling, it seems natural to be related.

Response:  We thank the reviewer for his useful comments. PASP and TAPSE were removed from the multiple linear regression analysis.

  • Was there a connection between SPAP and TAPSE? Interesting results may be seen by plotting the values ​​at baseline, 1 year and 2 years for each case, with the vertical axis representing SPAP and the horizontal axis representing TAPSE.

Response: We thank the reviewer for his helpful comments. We added a figure (Figure 3) with plotted values of SPAP and TAPSE for each case.

Reviewer 3 Report

The authors present a retrospective analysis of patients with HFrEF on Sacubitril/Valsartan treatment. They observed improvement of RV function parameters as well as TAPSE/PASP over a one year follow-up. Concluding therapeutic success based on retrospective analyses is always problematic and should be discussed more critically in the Limitations section.

Overall the concept is interesting, but several methodological aspects raise concerns whether the study data allow for the conclusions the authors draw.

My specific comments are:

  • With the rise of new pharmaceutical agents, It is important to take treatment effects of eg. SGLT-2 inhibitors into account. Given that almost 50% of your cohort had diabetes, I assume a substantial proportion received SGLT-2 inhibitors.
  • My biggest concern is that your signal results from an overall better patient management, including adequate diuretic treatment etc. In that regard, it would be important to mention change of medication during follow-up (increase of diuretics, start of MRA, start of SGLT-2i, etc).
  • It is not surprising that in Table 2 the two components of RV-PA coupling are predictors. Add additional data such as atrial size; this would somehow take diuretic treatment into account.
  • Include data on TR as this plays an important role when discussing RV function and pulmonary circulation pressures. Include TR also in your regression analyses. Does severity decrease during ARNI treatment?
  • In general, the Results section is only the two tables. You should elaborate specific items which you later discuss in your Discussion section. What about sex-specific differences, what about diabetic versus non-diabetic; ischemic/non ischemic. Right now it reads like a short research letter but not a full manuscript.
  • Please check for typos (eg abstract line1 “assested”)
  • In Tables, some units are missing for certain parameters (eg. Table 1. eGFR, NT-proBNP,…)
  • Discuss postcapillary pulmonary hypertension in HFrEF.

Author Response

The authors present a retrospective analysis of patients with HFrEF on Sacubitril/Valsartan treatment. They observed improvement of RV function parameters as well as TAPSE/PASP over a one-year follow-up. Concluding therapeutic success based on retrospective analyses is always problematic and should be discussed more critically in the Limitations section.

Response: We thank the reviewer for his valuable comment. We added the Limitations section that contains a critical discussion on our study.

Overall the concept is interesting, but several methodological aspects raise concerns whether the study data allow for the conclusions the authors draw.

My specific comments are:

  • With the rise of new pharmaceutical agents, It is important to take treatment effects of eg. SGLT-2 inhibitors into account. Given that almost 50% of your cohort had diabetes, I assume a substantial proportion received SGLT-2 inhibitors.

Response: Thank you for your useful suggestions. However, we do not believe that treatment with SGLT-2 may have affected the results of our study. In fact, only 34 (20.8%) diabetic patients included in the study were treated with SGLT-2 inhibitors. In Italy, these drugs are only prescribed by a diabetologist and cannot be prescribed in patients with eGFR < 60 ml/min (in case of the first prescription) or with eGFR < 45 ml/min (in case of continued treatment).

  • My biggest concern is that your signal results from an overall better patient management, including adequate diuretic treatment etc. In that regard, it would be important to mention change of medication during follow-up (increase of diuretics, start of MRA, start of SGLT-2i, etc).

Thank you for pointing this out. However, we kindly disagree with it. During the follow-up, there has not been a significant increase in the administration of disease-modifying drugs or in the percentage of patients treated with MRA. With regard to the treatment with diuretics, the introduction of sacubitril/valsartan determines, as documented both in clinical trials and observational studies, the need to reduce the dose of diuretics and not to increase it; otherwise, hypovolemia will occur. During the follow-up period, the dose of diuretic agents administered to our study patients was reduced.

  • It is not surprising that in Table 2 the two components of RV-PA coupling are predictors. Add additional data such as atrial size; this would somehow take diuretic treatment into account.

Response:  We thank the reviewer for his useful comment. We removed PASP and TAPSE from multiple linear regression analysis and added LAVi.

  • Include data on TR as this plays an important role when discussing RV function and pulmonary circulation pressures. Include TR also in your regression analyses. Does severity decrease during ARNI treatment?

Response: We agree with the reviewer that TR plays an important role in RV function and pulmonary circulation. Of course, TR severity decreases after initiation of sacubitril/valsartan treatment; however, as TR was evaluated qualitatively and not quantitatively, we cannot include it in the regression analysis.

  • In general, the Results section is only the two tables. You should elaborate specific items which you later discuss in your Discussion section. What about sex-specific differences, what about diabetic versus non-diabetic; ischemic/non ischemic. Right now it reads like a short research letter but not a full manuscript.

Response:  We thank the reviewer for his useful comment. The Discussion section was expanded further according to the reviewer’s suggestion

  • Please check for typos (eg abstract line1 “assested”)

Thank you for your helpful comment. All typographical errors have already been corrected in accordance with the reviewer’s suggestion.

In Tables, some units are missing for certain parameters (eg. Table 1. eGFR, NT-proBNP,…)

Thank you for pointing this out. The missing units for certain parameters have already been provided.

  • Discuss postcapillary pulmonary hypertension in HFrEF.

We thank the reviewer for his useful comment. A statement regarding the role of type 2 pulmonary hypertension in HFrEF has already been added in the Discussion section.

Round 2

Reviewer 3 Report

The authors do highlight which sections they changed in the revised manuscripts which makes it very hard for reviewers. 

Your response regarding the SGLT2 inhibitors is insufficient and should be discussed in the manuscript. Overall, I recommend taking comments by reviewers serious as it is frustrating to realise that no action is taken for most comments. 

"Thank you for pointing this out. However, we kindly disagree with it. During the follow-up, there has not been a significant increase in the administration of disease-modifying drugs or in the percentage of patients treated with MRA." Please provide data for your statement. 

Author Response

We are sorry that the reviewer misunderstood our response and apologize to the reviewers and editors.
We have taken the reviewer's comments very seriously and have made the required changes to the best of our ability.
In the revised manuscript the changes were indicated by the track system
changes" function of Microsoft Word.
In order to avoid further misunderstandings, we have put in bold the modified parts according to the comments of the editor
As a demonstration of how seriously we took the words of the reviewer who indicated the presence of some grammatical errors in the manuscript, we also sent the paper for a grammatical revision to Enago.
We apologize again for the misunderstanding and wish the editor good work.